# Novel Somatic Copy Number Alteration Identified for Cervical Cancer in the Mexican American Population

**DOI:** 10.3390/medsci4030012

**Published:** 2016-08-02

**Authors:** Alireza Torabi, Javier Ordonez, Brenda Bin Su, Laura Palmer, Chunxiang Mao, Katherine E. Lara, Lewis P. Rubin, Chun Xu

**Affiliations:** 1Department of Pathology, TTUHSC, El Paso 79905, TX, USA; alireza.torabi@ttuhsc.edu; 2Department of Biomedical Science, TTUHSC, El Paso 79905, TX, USA; jaordonez2003@gmail.com; 3Department of Pediatrics, Texas Tech University Health Sciences Center (TTUHSC), El Paso 79905, TX, USA; laura.palmer@ttuhsc.edu (L.P.); frankmao2007@gmail.com (C.M.); kelara@miners.utep.edu (K.E.L.); lewis.rubin@ttuhsc.edu (L.P.R.); 4Department of Internal Medicine, College of Medicine and Health Sciences, UAE University, Al-Ain 15551, UAE; brendabinsu@uaeu.ac.ae

**Keywords:** cervical cancer, copy number alterations, genome wide analysis, somatic, Mexican American population, Illumina HumanOmni2.5-8 BeadChip

## Abstract

Cervical cancer affects millions of Americans, but the rate for cervical cancer in the Mexican American is approximately twice that for non-Mexican Americans. The etiologies of cervical cancer are still not fully understood. A number of somatic mutations, including several copy number alterations (CNAs), have been identified in the pathogenesis of cervical carcinomas in non-Mexican Americans. Thus, the purpose of this study was to investigate CNAs in association with cervical cancer in the Mexican American population. We conducted a pilot study of genome-wide CNA analysis using 2.5 million markers in four diagnostic groups: reference (*n* = 125), low grade dysplasia (cervical intraepithelial neoplasia (CIN)-I, *n* = 4), high grade dysplasia (CIN-II and -III, *n* = 5) and invasive carcinoma (squamous cell carcinoma (SCC), *n* = 5) followed by data analyses using Partek. We observed a statistically-significant difference of CNA burden between case and reference groups of different sizes (>100 kb, 10–100 kb and 1–10 kb) of CNAs that included deletions and amplifications, e.g., a statistically-significant difference of >100 kb deletions was observed between the reference (6.6%) and pre-cancer and cancer (91.3%) groups. Recurrent aberrations of 98 CNA regions were also identified in cases only. However, none of the CNAs have an impact on cancer progression. A total of 32 CNA regions identified contained tumor suppressor genes and oncogenes. Moreover, the pathway analysis revealed endometrial cancer and estrogen signaling pathways associated with this cancer (*p* < 0.05) using Kyoto Encyclopedia of Genes and Genomes (KEGG). This is the first report of CNAs identified for cervical cancer in the U.S. Latino population using high density markers. We are aware of the small sample size in the study. Thus, additional studies with a larger sample are needed to confirm the current findings.

## 1. Introduction

Based on the U.S. Centers for Disease Control and Prevention (CDC) 2009 report, 12,357 women were diagnosed with cervical cancer, and 3909 women died from cervical cancer in the United States. This huge number of patients with cervical cancer has been a critical issue and is responsible for 10%–15% of cancer-related deaths in females globally [1]. Cervical cancer is the second most common malignant tumor in women worldwide. Among these tumors, approximately 80% are squamous cell carcinomas (SCCs), and 5%–20% are adenocarcinomas (AdCAs) [2,3]. Cervical SCCs are developed from a premalignant disease known as cervical intraepithelial neoplasia (CIN) Graded 1–3 with increasing atypical features. The five-year overall survival for cervical cancer is only 66%.

In addition, increasing evidence suggests that infection with high-risk subtypes of human papillomavirus (HPV) (e.g., HPV-16 and HPV-18) is the most common cause and is the primary initiator of premalignant lesions [4]. However, only a small proportion of women infected with oncogenic HPV subtypes develop cervical cancer, which suggests that HPV infection alone is insufficient to cause cancer, and there is a possibility of other host factors linked to the development of invasive cervical cancer [5], like genetic variation, including polymorphisms, insertions or deletions in the host genome [6,7]. Increasing evidence demonstrates that there was a consistent relationship between certain genetic variants (such as the tumor protein 53 (TP53) Arg72Pro polymorphism) and cervical cancer, most likely modulated by the presence of high-risk HPV during progression from squamous intraepithelial lesions (SIL) to cervical cancer.

Moreover, Mexican American women in Texas have among the highest rates of cervical cancer incidence and mortality in the country [8] with twice the frequency as compared to their non-Mexican American counterparts. The annual death rate from cervical cancer for Mexican Americans is 24.2 out of 100,000 [9]. However, there is a lack of studies on somatic mutation identification for cervical cancer in this population. Only four studies have reported on genetic basis of cervical cancer in Mexican American women based on a PubMed search (19 November 2015). The number of the Mexican American population was estimated at 50.5 million in the 2010 census, making Mexican American the largest minority group in the U.S., as well as a rapidly growing segment of the U.S. population.

Recent advances in genome studies have led to the discovery of one important type of variation that can be assessed with recent technology: copy number alterations (CNAs), usually for a cancer study, somatic copy number changes and/or copy number variations (CNVs), usually for a non-cancer study, or germline copy number changes, such as CNVs identified for neuropsychiatric disorders in our recent study [10]. These CNAs or CNVs are by definition chromosomal regions with sizes of 1 kb to several Mb, which vary across individuals with regard to the number of copies of a chromosomal segment. CNVs refer to structural variations of the DNA that include insertions, deletions and duplications. Studies have found that CNVs cover as much as 14% of the human genome [11], and there is a much higher de novo rate as many as 10–1000-fold in CNVs as compared to single nucleotide polymorphisms (SNPs) [11,12]. Furthermore, CNVs have been shown to account for more genomic differences between individuals than SNPs [13,14]. Therefore, CNVs may contribute a sizeable amount of disease phenotypic variation in each individual of a population [15].

Analyses integrating mutation information with data on rearrangements and CNAs have revealed a higher-order organization of the seemingly random genetic events that lead to cancer [16]. Interestingly, genes in regions subject to copy number changes appear to be organized along functional ontological terms related to cancer [16]. Studies have also implicated a number of somatic mutations, including *TP53*, phosphatidylinositol-4,5-bisphosphate 3-kinase catalytic subunit alpha (*PIK3CA*), phosphatase and tensin homolog (*PTEN*), serine/threonine kinase 11 (STK11) and V-Ki-ras2 Kirsten rat sarcoma viral oncogene homolog (*KRAS*) [17,18], and several CNAs in the pathogenesis of cervical carcinomas [19,20] in non-Mexican Americans.

In this study, we carried out a genome-wide survey of potential somatic CNAs, including amplifications and deletions, in apparently normal tissues (*n* = 2), low grade dysplasia (*n* = 4), high grade dysplasia (*n* = 5), invasive carcinoma (*n* = 5) and blood samples (*n* = 125, serving as a reference group) of female subjects from the HapMap data. We are aware of the limited number of cases and the lack of a control group. Thus, in the future, a large study with a control sample and more cases as a methodological alternative is needed. We genotyped 2.5 million markers and analyzed somatic CNAs in a total of 14 tissues using the Illumina HumanOmni2.5-8 BeadChip Kit at tissue-level resolution. We mapped genomic changes: (1) between peripheral blood samples of the reference subjects and cervical tissues from the cases (cervical dysplasia and invasive carcinoma); because only two normal cervical tissue samples had insufficient statistical power, we excluded these two samples from the further analysis; (2) we also analyzed genomic changes among four diagnostic groups (normal, low, high grade and invasive carcinoma). We expect that this study: (i) will provide an estimate of the prevalence of somatic CNAs by identifying specific patterns, genes and/or biological pathways associated with different stages of cervical dysplasia in Mexican Americans; (ii) will investigate the genomic context of these somatic CNAs; and (iii) will evaluate whether the burden of somatic mutations predicts tumor progression.

## 2. Materials and Methods

### 2.1. Materials

A total of 14 tissues (low, high grade dysplasia and invasive carcinoma) from cases and 125 female subjects, serving as a reference group, were used for this study. The demographic information is shown in Table 1. The cases were categorized into three groups, including low grade dysplasia (CIN-I, *n* = 4), high grade dysplasia (CIN-II and III, *n* = 5) and squamous cell carcinoma (SCC, *n* = 5) groups. For a broader definition of the case group, we also divided the cases into two groups, pre-cancer (CIN-I, -II and -III) and cancer (SCC). All of the case subjects in the current study were from the Mexican American population recruited from the outpatient clinics at the University Medical Center (UMC) and Texas Tech University Health Sciences Center (TTUHSC)-El Paso. All cases of cervical cancer were diagnosed as SCC by histopathological examinations, whereas healthy women had no abnormal cytological findings in the Pap smear tests of the uterine cervix.

The Illumina HumanOmni2.5-8 BeadChip data from a total of 125 blood samples of female subjects from the HapMap were used as a reference in the current study.

### 2.2. Methods

#### 2.2.1. Tissue Specimens

Sixteen cervical tissue samples were obtained from the Department of Pathology, TTUHSC-El Paso. We excluded two normal tissues due to an insufficient tissue sample size. All case subjects were HPV positive, from the Mexican American population and had signed Institutional Review Board-approved written informed consent forms prior to enrolling in the study. The procedures were approved by the Institutional Review Boards of TTUHSC (IRB #E13107), and the study was performed in accordance with the Helsinki Declaration of 1975.

#### 2.2.2. Microdissection

Paraffin-embedded tissues were first sectioned into 10-μm slices, which were hematoxylin-eosin stained for the selection of the appropriate tissue area. The corresponding selected areas of each tissue sample were then collected under microscopic observation with a 30-gauge needle (Becton-Dickinson, Franklin Lakes, NJ, USA).

Genomic DNA of micro-dissected tissue was extracted by proteinase K digestion followed by standard phenol-chloroform extraction. The QIAamp Formalin-Fixed, Paraffin-Embedded (FFPE) Tissue Kit from Qiagen (Valencia, CA, USA), which is widely used for extracting DNA from FFPE sections, was used. The experiment was performed according to the manufacturer’s handbook. The total amount of DNA was spectrophotometrically determined by measuring the absorbance at 260 nm (A260), and DNA purity was assessed by detecting the A260/A280 ratio using the Varioskan Flash (Thermo Scientific, Rockford, IL, USA) according to the manufacturer’s instructions.

#### 2.2.3. High Density Genotyping

Genomic DNA from the case group was used to obtain genotypes by the Illumina HumanOmni2.5-8 BeadChip (Illumina, San Diego, CA, USA). This DNA chip provided over 2.5 million markers at a median spacing of 1.2 kb and full support of CNV or CNA applications, which was a powerful genotyping tool and allowed us to make more meaningful discoveries. Genomic annotations were based on National Center for Biotechnology Information (NCBI) Human Genome Build 37 (University of California, Santa Cruz Genome Browser Release 19), and genotyping experiments were performed at the Genomics Core at the TTUHSC-El Paso.

Due to our limited number of control samples of cervical tissues, we used publicly available HapMap data with genotypes, including CNV data of the same DNA chip, the Illumina HumanOmni2.5-8 BeadChip Kit, in the 125 female subjects as a reference group. The HapMap data with genotypes were downloaded from [21].

In addition, there is no genotype data available for the Mexican American population using the HapMap data, so we selected these 125 female subjects from the Admix populations (Utah residents with Northern and Western European ancestry, CEU; Yoruba, YRI; Han Chinese, CHB, and Japanese, JPT) since the various combinations of reference panels with the multi-ethnic populations had better accuracy than those containing only single ethnic samples [22] for genetic study and genetic imputation analysis.

The raw genotyping signal data were processed by the Illumina GenomeStudio software Genotyping Module Version 3.2.33 (Illumina) and converted to allele-specific intensity values. The genotype call rate threshold was set at ≥95% for all samples, and a total of 139 including case (*n* = 14) and reference (*n* = 125) subjects passed the quality control. A Partek customized report of the normalized genotype data, composed of those 139 samples, including low grade (CIN-I = 4), high grade (CIN-II and CIN-III = 5), SCC (*n* = 5) and 125 females from the reference group (Table 1), was transferred to Partek^®^ Genomic Suite^®^ software, Version 6.6, Copyright 2014, Partek Inc. (Saint Louis, MO, USA), for downstream analysis.

#### 2.2.4. CNA Detection

Unpaired copy number analysis was performed in the Partek Genomics Suite comparing allele intensities to a reference baseline of 125 female HapMap admixed samples using a similar analysis strategy as a previous publication [23]. The genomic segmentation algorithm was applied to find break points and to detect amplifications (gains) and deletions (losses). The following stringent parameters were used to identify CNAs and CNA regions as a previous study [24]: (1) each segment must contain a minimum of 10 consecutive filtered probe sets; (2) a *p* value threshold of 0.001 when compared to the neighboring adjacent regions; and (3) a signal-to-noise threshold of 0.5 and diploid copy number 1.7 to 2.3.

#### 2.2.5. Pathway Analysis

We further examined whether these CNAs in the various genes have an impact on gene functions. Gene ontology (GO) analysis was also performed using the Partek^®^ Genomic Suite^®^ software, Version 6.6, Copyright^®^ 2014, Partek Inc., to investigate whether there was enrichment for cancer-associated CNAs (*p* < 0.05) in genes from any ontology categories.

Disease association of individual CNA frequencies with patients, as a group, compared to a reference group, was assessed using 2 × 2 or 2 × 3 contingency tables, two-tailed χ^2^ tests or Fisher exact tests. Statistical analyses were performed using the SPSS statistical software package (Version 10, IBM, Chicago, IL, USA). Differences with two-tailed probability values of *p* ≤ 0.05 were taken as statistically significant. *p*-values for tests of CNA association were conservatively corrected for multiple testing using the Bonferroni method.

## 3. Results

### 3.1. Overall CNA Patterns

We processed genotype and intensity data for all 2.5 million probes of the Illumina HumanOmni2.5-8 BeadChip Kit for the 14 cases and 125 reference subjects. A number of CNAs (deletions or losses) was present in many chromosomal regions (Figure 1). A total of 2220 CNAs >1000 bp was identified, including 725 (32.7%) amplifications (gains) and 1495 (67.3%) deletions (losses) mainly on the 22 autosomal chromosomes.

The chromosomal locations of the copy number amplifications and deletions of the 22 autosomes and X chromosomes were shown by karyograms (Figure 1). Most of the amplifications are found in the cases at the short arms of chromosomes 1, 5, 8, 16, 19 and 20, as well as the centromeres of chromosomes 14, 15 and 21. Most of the deletions in the cases were mainly observed at the short arms of chromosomes 1, 4, 5, 7, 11, 12, 16, 17 and 19, as well as the centromeres of chromosome 19.

### 3.2. Principal Component Analysis (PCA)

To characterize aberration profiles in different diagnostic groups (low grade dysplasia, high grade dysplasia, SCC and reference), we performed a PCA (Figure 2). There was a great variability in the three diagnostic groups (reference, pre-cancer and cancer) and four diagnostic groups (reference, low grade, high grade and SCC). These results indicate that there is a distinct difference in the patterns of the CNAs between case and reference groups based on the genomic profiles of these subjects.

Overall, there was a clear separation between the reference and the case group when examined by PCA clustering.

### 3.3. Genome-Wide CNA and Cancer Progression

Increasing evidence demonstrates that more numerical genomic alterations are associated with progression from precursor lesions to invasive cancer [25]. Thus, we conducted CNA genomic analysis in different diagnostic groups to understand the complexities of the genomic architecture of these highly heterogeneous groups of cervical cancer and to assess whether CNA burden impacts the prognosis of cervical cancer in this Mexican American population. We analyzed CNAs among two (pre-cancer and cancer; Table 2B) and three diagnostic groups (low grade, high grade and SCC; Table 2A) in cases only. There was no statistically-significant difference of CNA burden among diagnostic groups in cases in any size of CNA, although there are slightly higher frequencies of amplifications in the 100 kb, 10–100 kb and 1–10 kb categories in the cancer group (11.0%, 23.3% and 13.4%, respectively) as compared to the pre-cancer group (7.5%, 18.5% and 12.3%, respectively) (Table 2) using 205 two-tailed χ^2^ tests with the SPSS statistical software package.

### 3.4. Recurrent Aberrations Identified in Cases Only

We are interested in recurrent CNAs since the chromosome regions with recurrent CNAs more likely to harbor disease-critical genes are those that show alterations that are recurrent among individuals with cancer or other diseases [26,27]. In this context, we can define a recurrent CNA region as a set of contiguous genes (a region) that shows a high enough probability (or evidence) of being altered (e.g., deletion or insertion) in two or more samples, as previously described [28]. The recurrent aberrations in a total of 98 CNA regions containing 849 genes/loci were identified only in the case group (pre-cancer and cancer subjects) with at least two cases who carried these CNAs. None of these CNAs were observed in the reference group (Table 3). Among these 98 recurrent CNA regions, there were 545 deletions (84%) and 104 amplifications (16%) observed in cases only using statistical analyses described in the Methods section.

Table 3 shows the top 20 chromosome regions with a high number of recurrent somatic CNAs (>100 kb) in the cases with at least nine subjects carrying these deletions in each CNA region; none of these CNAs were observed in the reference group. Some of the CNAs contain tumor suppressor genes (e.g., axis inhibition protein 1 (*AXIN1)* and tuberous sclerosis complex 2 (*TSC2)*).

Eight exceptionally large CNA regions with deletions of between 1591 kb and 517 kb were detected in cases on short arms of chromosomes 1, 16 and 19.

In addition to recurrent deletions identified in the cases (Table 3), two recurrent CNA regions with a 220-kb amplification on 21q11.2 and a 113.5-kb deletion on 7p22.2 were observed in only CIN-III and SCC.

Next, we checked the Cervical Cancer Database (CCDB) [29] and found three genes, haemoglobin alpha 2 (*HBA2*), mesothelin (*MSLN*) and *STK11*, overlapped between 849 genes/loci observed from our current study and 538 genes listed in the CCDB.

### 3.5. CNA Regions Contained Known Tumor Suppressor Genes and Oncogenes in Cancer Tissues but Not in the Reference Group

We were also interested in identifying CNA regions that were observed only in cases and over 100 kb containing known cancer-related genes as compared to the references (Table 3, Table 4 and Table 5). Thus, we examined whether newly-identified CNAs contained tumor suppressor genes and/or oncogenes. We observed a total of 26 CNA regions containing known tumor suppressor gene regions. These CNAs were deletions and only seen in the case group. Furthermore, none of these regions were observed in the reference group (Table 4). We also observed a total of six CNA regions with deletions (no amplification) containing known oncogenes in the case group, and none of these regions were observed in the reference group (Table 5).

### 3.6. Pathway Analysis

Recent studies have incorporated protein networks into the results of genome-wide CNA data using networks or GO analysis to discover disease-associated and/or enriched pathways [30,31,32,33]. Therefore, we conducted GO analysis and Kyoto Encyclopedia of Genes and Genomes (KEGG) pathway analysis.

The results of the functional enrichment of KEGG pathway analysis in the identified CNAs >100 kb that occurred in two or more cases are shown in Table 6. Fourteen pathways were discovered, including the insulin signaling pathway and the endometrial cancer and estrogen signaling pathway, with enrichment *p* values lower than 0.05. A GO analysis was performed by selecting CNAs in the genes to see if any GO categories were overrepresented among CNAs identified in cases [34]. Many biological processes were undisturbed at the molecular level, while others were frequently affected across multiple cases, as shown in Table 7, which lists the most commonly-affected function with *p*-values lower than 0.05 and more than four genes in each function category.

## 4. Discussion

It has been demonstrated that somatic structural alterations (e.g., amplifications or deletions) of human chromosomes represent a common class of mutations, which may cause gene disruption (e.g., deletion or rearrangement), gene activation (e.g., CNAs, gain or amplification) or the formation of novel oncogenic gene products (gene fusions). Many of these events actively drive carcinogenesis [35,36]. In our initial cervical cancer cohort, some CNA patterns identified were previously found to be correlated with cervical cancer.

We screened whole genomes with 2.5 million markers of an array to discover any recurrent copy number alterations in cases (pre-cancer and SCC). We identified a total of 98 CNA regions >100 kb in the case group, including low grade dysplasia, high grade dysplasia and SCC. These CNAs occurred in two or more cases, including two large deletions (1591 kb and 1123 kb) (Table 3). Of the top 20 CNA regions >100 kb deletions in cases only, six CNAs occurred on 16p13 that have been reported in cervical cancer [37], and mutation analysis of the *AXIN1* gene located at 16p13 was reported to be involved in the Wnt pathway in cervical carcinomas [38]. The CNA containing the SET and MYND domain-containing protein 4 (*SMYD4*) gene on 17p13.3 was also reported previously in this cancer [37]. The *SMYD4* gene is demonstrated to be a potential tumor suppressor and plays a critical role in breast carcinogenesis, at least partly through inhibiting the expression of platelet-derived growth factor receptor-alpha (*PDGFRA*), and this gene could be a novel target for improving the treatment of breast cancer [39].

A number of previous studies demonstrate that numerical chromosomal aberrations were found to progress to invasive cancer from precursor lesions [25,37]; however, we did not observe this feature in our current study sample. This might be due to the small sample size or any other unknown or not yet identified factors, such as ethnicity, since most previous study populations are non-Hispanic populations.

Among two amplifications identified in CIN-III and SCC, one recurrent CNA region with a 220-kb amplification on 21q11.2 was observed in CIN-III and co-occurred with previous findings, where the amplification had been identified in breast cancer subjects with tamoxifen resistance [40].

The recurrent CNA regions we identified contain a number of tumor suppressor genes, oncogenes and cancer-related genes. A CNA that contains the interferon-induced transmembrane protein 1 (*IFITM1*) gene at 11p15.5 was identified in eight cases. This gene has been reported to be involved in cervical carcinogenesis [41]. A total of nine cases with the CNA containing the fucosyltransferase 3 (*FUT3*) gene at 19p13.3 were observed in the current study, and this gene is associated with breast cancer [42]. A CNA region that includes the naked cuticle 2 (*NKD2*) gene at 5p15.35 was detected in six cases. This gene was found to suppress breast cancer proliferation by inhibiting Wnt signaling [43]. The ovarian cancer-associated gene 2 protein (OVCA2) at 17p13.3 was repeatedly reported to be associated with ovarian cancer, and we observed five cases carrying the CNA at this gene region. A cluster of three tumor suppressor genes, cadherin 1 (*CDH1*), death-associated protein kinase (*DAPK*) and hypermethylated in cancer 1 (*HIC1*), displayed a significantly increased frequency of promoter methylation with progressively more severe cervical neoplasia. In addition, the Hes family BHLH Transcription Factor 5 (*HES5)* gene was reported to be associated with cervical carcinoma cells using immunocytochemistry, Western blot and methyl thiazolyl tetrazolium assays [44], and the CNAs on 1p36.33-1p36.32 containing the *HES5* gene were also identified in the current study. Our newly-identified CNA containing this gene on 7p22.2 was also observed in the recurrent pre-cancer and cancer subjects (Table 3).

Moreover, we also examined deletion burden observed to be similar to those for other cancers and found more deletions identified than amplification in most of the cancer studies, which supports our findings. A recent study using the TCGA data identified nine regions of deletion that were unique to estrogen receptor positive (ER+) post menopause tumors in patients with breast cancer [45], including deletion in 7p22.3, where our newly-identified deletion in cases only was located, and it contains a known tumor suppressor gene.

To analyze the possible effect of genomic alterations, to further capture cancer-causing gene information and to see whether any GO categories are overrepresented among CNA regions, we searched the KEGG pathway database and GO categories and identified a number of pathways and functions where CNAs occur in the SCC group, but not in the reference group (Table 6 and Table 7). Furthermore, our results were partly in agreement with previous reports about cervical cancer. For example, using the functional enrichment of KEGG pathway analysis, we discovered the insulin signaling pathway in the case group, and it was evidenced that HPV 16 E6 oncoprotein interferes with the insulin signaling pathway by binding to tuberin [46]. As is already known, high-risk HPV infection is a causal agent for cervical cancer. In addition, two pathways of interest are endometrial cancer and the estrogen signaling pathway observed in the current study. The NF-κB signaling pathway was also identified in our cancer group (SCC), but not in the reference group, which supports a previous study where bisphenol A (BPA) stimulated the cervical cancer migration via IKK-β/NF-κB signals [47]. The estrogen signaling pathway was also shown in our KEGG analysis. A previous study was performed to evaluate the potential of miRNAs as novel markers for the post-therapeutic monitoring of cervical SCC patients. A regulatory network of differentially-expressed serum miRNAs was identified, and a number of target genes was predicted in the estrogen-mediated signal pathways [48]. Ours and others’ findings support that cervical cancer is a hormone-associated gynecological disease. For example, HPV infection has been associated with the deregulation of the PI3K-Akt-mTOR pathway in invasive cervical carcinomas [49]. The PI3K-Akt signaling pathway was also found in our current study, and 14 genes in this pathway were found in two or more SCC cases (Table 6).

Moreover, signal transducer activity with an enrichment score of 4.2 and a *p* value of 0.01 containing 11 genes was also identified in a previous report on cervical cancer [50] using GO analysis. Signal transduction was identified in our current study in cervical cancer (SCC), but not in the reference group; a recent study also supports that a signal transduction cascade and mitogen-activated protein kinase kinase kinase 3 (MEKK3) serve as key integration points and are important factors in regulating cellular responses to environmental stress. This signal transduction and MEKK3 not only link diverse extracellular stimuli to subsequent signaling molecules, but they also amplify the initiating signals to ultimately activate effector molecules and induce cell proliferation, differentiation and survival of cervical cancer [51].

These pathways and functions were identified as overlapping biological themes, and these data may provide useful information on the molecular mechanisms for cervical cancer, its prognosis and treatment responses.

There are a number of novelties in this study: (1) this is the first report of CNAs in the relatively ethnically homogeneous group of Mexican Americans using high density mapping (2.5 million in number); previous studies using less density markers may result in an underestimation of the genetic changes that take place in cervical cancer; (2) newly-identified CNAs in cases together with results in the in silico analysis using KEGG and GO function provide insight into how multiple CNAs might contribute to cervical cancer development. We also are aware of some limitations in our study. (1) Our small sample size (14 cases and 125 references) is a major limitation for this type of study. Due to a small number of samples, the nature of genome-wide alteration of copy number may not be fully explained in pre-cancer and SCC. Future confirmation studies using an independent sample will provide an opportunity to more accurately dissect the genetic complexity of somatic CNAs for cervical cancer. (2) There might be a bias of the CNA identified in cervical cancer for the Mexican Americans, since we used the 1000 Genome admixed populations, not Mexican Americans; thus, we currently are recruiting more subjects with cervical cancer from the same population and plan to validate the findings in more samples. (3) Other biomarkers, such as RNA-Seq or DNA methylation profile and sequencing data using next generation sequencing technologies (such as target gene sequencing in these CNA regions), will provide an opportunity for in-depth molecular profiling of the fundamental biological processes of cervical cancer.

We expect that with future validation and confirmation, newly-discovered CNAs can be used in cancer classification, diagnosis, prognosis and treatment responses.

## 5. Conclusions

Using high density SNP array analysis, we have shown extensive genome-wide CNA changes in pre-cancer and cancer groups as compared to the genome CNA profile in the reference group. Our results demonstrated that the recurrent alterations of CNAs occurred in cases of pre-cancer and SCC. Some of the somatic genomic gains and losses in cervical pre-cancer and cancer were in concordance with the results from previous studies. To our knowledge, no previous studies have applied genome-wide copy number analysis using such high density markers for cervical cancers in the Mexican American population; however, validation and confirmation in a larger sample size are needed.

## Figures and Tables

**Figure 1 medsci-04-00012-f001:**
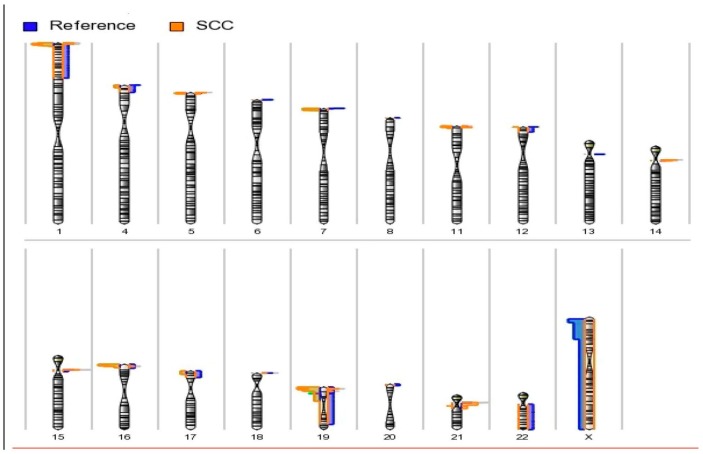
Karyogram view of detected amplified and deleted regions across autosomes. Amplifications are shown at right side of the chromosomes, and deletions are shown at the left side of the chromosomes. The length of the horizontal bar corresponds to the number of samples observed at the respective cytobands. Most of the amplifications were found in cases at the short arms of chromosomes 1, 5, 8, 16, 19 and 20, as well as the centromeres of chromosomes 14, 15 and 21. Most of the deletions in the cases were mainly observed at the short arms of chromosomes 1, 4, 5, 7, 11, 12, 16, 17 and 19, as well as the centromere of chromosome 19. We observed a statistically-significant difference of CNA burden between case and reference groups (Table 2) for different sizes of CNAs (>100 kb, 10–100 kb and 1–10 kb). For example, statistically-significant differences of >100 kb, 10–100 kb and 1–10 kb deletions were observed between reference (6.7%, 2.5%, 0.8%), pre-cancer (92.5%, 81.5%, 87.7%) and cancer (89.0%, 76.7%, 86.6%) groups, respectively (Table 2).

**Figure 2 medsci-04-00012-f002:**
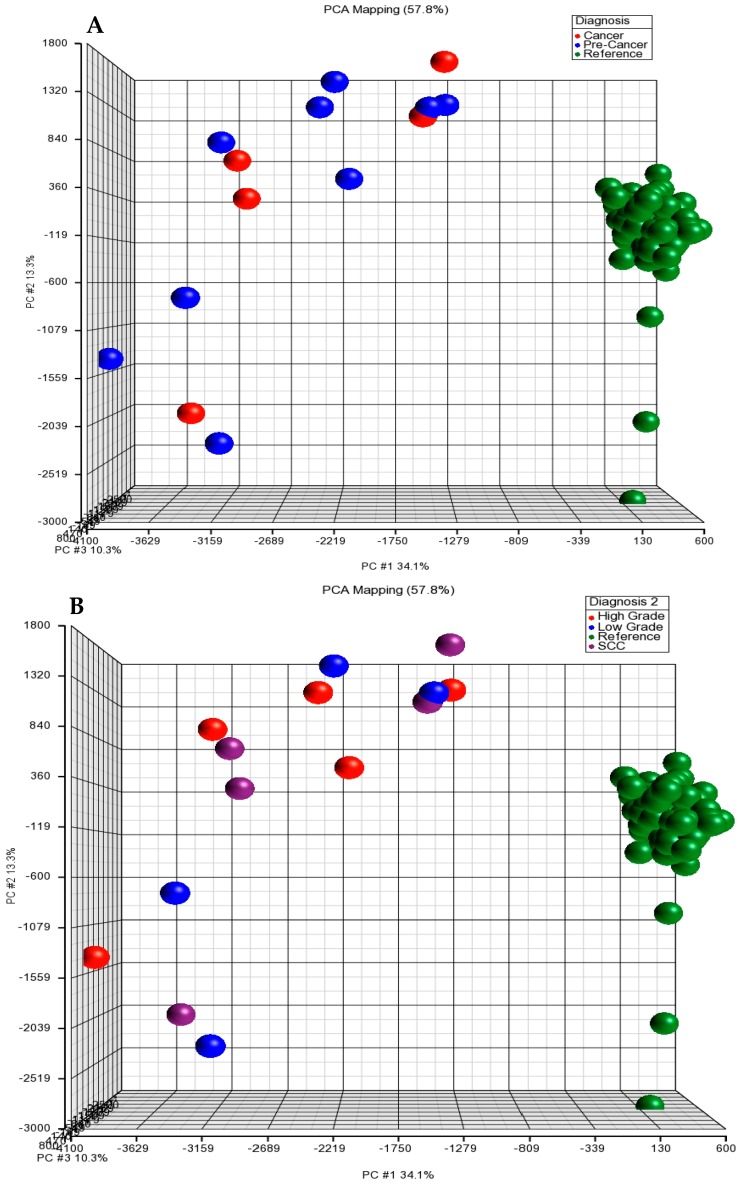
Plot of principal component analysis (PCA) and hierarchical clustering of copy number variation (CNV) or CNA datasets. (**A**) PCA scatter plot of three diagnostic groups (pre-cancer, cancer and reference). Each point represents a specific sample. Points are colored by group status, with blue representing pre-cancer (CIN-I, -II and -III), red representing invasive cancer and green representing references. (**B**) PCA scatter plot of four diagnostic groups. Each point represents a specific sample. Points are colored by group status, with blue representing low grade (CIN-I), red representing high grade (CIN- II and -III), purple representing invasive cancer and green representing references.

**Table 1 medsci-04-00012-t001:** Clinical demographics with diagnosis, age for cases and reference subjects.

Subject ID	Age (years)	Diagnosis 1	Diagnosis 2	Diagnosis 3
14	42	CIN-I	low grade dysplasia	pre-cancer
16	24	CIN-I	low grade dysplasia	pre-cancer
17	53	CIN-I	low grade dysplasia	pre-cancer
15	31	CIN-I	low grade dysplasia	pre-cancer
12	27	CIN-II	high grade dysplasia	pre-cancer
8	39	CIN-III	high grade dysplasia	pre-cancer
11	21	CIN-III	high grade dysplasia	pre-cancer
6	30	CIN-III	high grade dysplasia	pre-cancer
5	38	CIN-III	high grade dysplasia	pre-cancer
3	55	SCC	SCC	SCC
4	44	SCC	SCC	SCC
2	51	SCC	SCC	SCC
7	31	SCC	SCC	SCC
1	42	SCC	SCC	SCC
*n* = 125 females	NA	reference	reference	reference

All subjects in the cases were human papillomavirus (HPV) positive and from the Mexican American population. Reference subjects were from an admixed population. NA, not available. CIN, cervical intraepithelial neoplasia; SCC, squamous cell carcinoma.

**Table 2 medsci-04-00012-t002:** Copy number alteration (CNA) burden (deletion and amplification) among different diagnostic groups.

**2 A**												
**Size of CNAs**	**Reference (*n* = 122)**			**Low (*n* = 4)**			**High (*n* = 5)**			**SCC (*n* = 5)**		
	All	Amplification (%)	Deletion (%)	All	Amplification (%)	Deletion (%)	All	Amplification (%)	Deletion (%)	All	Amplification (%)	Deletion (%)
>100 kb	212	198 (93.3)	14 (6.7)	156	15 (9.6)	141 (90.4)	244	15 (6.1)	229 (93.9)	218	24 (11)	194 (89)
10–100 kb	157	153 (97.5)	4 (2.5)	119	24 (20.2)	95 (79.8)	183	32 (17.5)	151 (82.5)	163	38 (23.3)	125 (76.7)
1 kb–10 kb	128	127 (99.2)	1 (0.8)	125	11 (8.8)	114 (91.2)	193	28 (14.5)	165 (85.5)	179	24 (13.4)	155 (86.6)
**2 B**												
**Size of CNAs**	**Reference**			**Pre-Cancer (*n* = 9)**			**Cancer (*n* = 5)**					
	All	Amplification (%)	Deletion (%)	All	Amplification (%)	Deletion (%)	All		Amplification (%)		Deletion (%)	
>100 kb	212	198 (93.3)	14 (6.7)	400	30 (7.5)	370 (92.5)	218		24 (11)		194 (89.0)	
10–100 kb	157	153 (97.5)	4 (2.5)	302	56 (18.5)	246 (81.5)	163		38 (23.3)		125 (76.7)	
1 kb–10 kb	128	127 (99.2)	1 (0.8)	318	39 (12.3)	279 (87.7)	179		24 (13.4)		155 (86.6)	

**Table 3 medsci-04-00012-t003:** Top 20 CNA chromosome regions with recurrent somatic CNAs (deletions >100 kb) in the cases; not observed in the reference group.

Cytoband	Start	Stop	# of Sub. with Del*.	Deletions in Subjects	No. of Markers	Length (bp)	Cancer-Related Genes
19p13.3	766,463	2,357,625	10	I (2), II (1), III (3), SCC (4)	1497	1,591,163	*STK11*
19p13.3	2,860,429	3,984,025	10	I (2), II (1), III (3), SCC (4)	1154	1,123,597	*MATK, DAPK3*
16p13.3	895,986	1,883,511	10	I (2), II (1), III (3), SCC (4)	1022	987,526	*AXIN1*
19p13.3	3,984,025	4,738,536	9	I (2), II (1), III (3), SCC (3)	672	754,512	
1p36.33-1p36.32	1,825,510	2,545,368	9	I (2), II (0), III (3), SCC (4)	732	719,859	*HES5*
1p36.32	2,545,368	3,230,848	9	I (2), II (0), III (3), SCC (4)	678	685,481	*PRDM16*
16p13.3	377,287	895,986	10	I (2), II (1), III (3), SCC (4)	525	518,700	*AXIN1, MSLN*
1p36.33	1,282,654	1,799,975	9	I (2), II (0), III (3), SCC (4)	242	517,322	
16p13.3	2,816,699	3,271,935	8	I (2), II (0), III (3), SCC (3)	469	455,237	*BSG, FGF22, AXIN1*
7p22.3	93,965	534,503	8	I (2), II (0), III (3), SCC (3)	265	440,539	
19p13.3	366,342	762,497	10	I (2), II (1), III (3), SCC (4)	604	396,156	*HBA2*
16p13.3	2,408,963	2,706,379	8	I (2), II (0), III (3), SCC (3)	142	297,417	
16p13.3	84,130	377,287	10	I (2), II (1), III (3), SCC (4)	284	293,158	*AXIN1*
16p13.3	2,132,651	2,397,391	10	I (2), II (1), III (3), SCC (4)	141	264,741	*TSC2*
1p36.33	1,021,380	1,282,654	9	I (2), II (0), III (3), SCC (4)	233	261,275	
16p13.3	1,888,316	2,115,231	10	I (2), II (1), III (3), SCC (4)	210	226,916	
1p36.33	810,836	1,021,380	9	I (2), II (0), III (3), SCC (4)	221	210,545	
19p13.3	2,532,493	2,735,499	10	I (2), II (1), III (3), SCC (4)	228	203,007	
19p13.3	2,367,858	2,532,493	10	I (2), II (1), III (3), SCC (4)	191	164,636	
19p13.3	254,506	365,334	10	I (2), II (1), III (3), SCC (4)	108	110,829	

# of Sub. with Del *: number of subjects with deletion. Serine/threonine kinase 11 (*STK11*); maturase K (*MATK*); Death-associated protein kinase 3 (*DAPK3*); axis inhibition protein 1 (*AXIN1*); hairy/enhancer of split, drosophila, homolog of, 5 (*HES5*); PR domain containing 16 (PRDM16); mesothelin (*MSLN*); basignin (*BSG*); fibroblast growth factor 22 (*FGF22*); haemoglobin A2 (*HBA2*); tuberous sclerosis complex 2 (*TSC2*).

**Table 4 medsci-04-00012-t004:** Newly-identified CNAs (deletions) contain known oncogenes in cases only.

Query Position	Gene Name	Gene ID	Length (bp)	Deletions in Different Diagnostic Groups
chr4:1694786–1937978	*WHSC1*	7468	243,193	ref (0), I (1), II (0), III (3), SCC (1)
chr4:1937978–2141625	*WHSC1*	7468	203,648	ref (0), I (1), II (0), III (3), SCC (1)
chr19:6194240–6239714	*MLLT1*	4298	45,475	ref (0), I (2), II (0), III (2), SCC (1)
chr19:6239714–6598484	*MLLT1*	4298	358,771	ref (0), I (2), II (0), III (2), SCC (1)
chr19:6679537–6796128	*VAV1*	7409	116,592	ref (0), I (2), II (0), III (1), SCC (1)
chr19:6796128–6883617	*VAV1*	7409	87,490	ref (0), I (2), II (0), III (1), SCC (1)

Wolf-Hirschhorn syndrome candidate 1 (*WHSC1*); myeloid/lymphoid leukemia; translocated to 1 (*MLLT1*); vav guanine nucleotide exchange factor 1 (*VAV1*).

**Table 5 medsci-04-00012-t005:** Newly-identified CNAs (deletions) contain known tumor suppressor genes in cases only.

Query Position	Gene Name	Gene ID	Length (bp)	Deletions in Different Diagnostic Groups
chr1:3252272–3613519	*TP73*	7161	361,248	ref (0), I (1), II (0), III (3), SCC (3)
chr1:3613519–3615102	*TP73*	7161	1584	ref (0), I (1), II (0), III (3), SCC (3)
chr1:3615102–3615341	*TP73*	7161	240	ref (0), I (1), II (0), III (3), SCC (3)
chr1:3615341–3621322	*TP73*	7161	5982	ref (0), I (1), II (0), III (3), SCC (3)
chr1:3621322–4125280	*TP73*	7161	503,959	ref (0), I (1), II (0), III (2), SCC (2)
chr5:204517–422812	*AHRR*	57491	218,296	ref (0), I (1), II (0), III (3), SCC (2)
chr5:422812–582058	*AHRR*	57491	159,247	ref (0), I (1), II (0), III (3), SCC (2)
chr7:1216332–1963055	*MAD1L1*	8379	746,724	ref (0), I (2), II (0), III (3), SCC (3)
chr7:1963055–1964889	*MAD1L1*	8379	1835	ref (0), I (2), II (0), III (3), SCC (3)
chr7:1964889–2219927	*MAD1L1*	8379	255,039	ref (0), I (2), II (0), III (3), SCC (3)
chr7:2219927–2770642	*MAD1L1*	8379	550,716	ref (0), I (2), II (0), III (3), SCC (3)
chr16:1888316–2115231	*TSC2*	7249	226,916	ref (0), I (2), II (1), III (3), SCC (4)
chr16:2115231–2116240	*TSC2*	7249	1010	ref (0), I (2), II (1), III (3), SCC (4)
chr16:2116240–2127184	*TSC2*	7249	10,945	ref (0), I (2), II (1), III (3), SCC (4)
chr16:2127184–2132651	*TSC2*	7249	5468	ref (0), I (2), II (1), III (3), SCC (4)
chr16:2132651–2397391	*TSC2*	7249	264,741	ref (0), I (2), II (1), III (3), SCC (4)
chr16:377287–895986	*AXIN1*	8312	518,700	ref (0), I (2), II (1), III (3), SCC (4)
chr16:84130–377287	*AXIN1*	8312	293,158	ref (0), I (2), II (1), III (3), SCC (4)
chr17:1586256–1692127	*SMYD4*	114826	105,872	ref (0), I (1), II (0), III (3), SCC (1)
chr17:1692127–1737406	*SMYD4*	114826	45,280	ref (0), I (1), II (0), III (3), SCC (1)
chr19:5310257–5598484	*SAFB2*	9667	288,228	ref (0), I (2), II (0), III (3), SCC (3)
chr19:5598484–5623377	*SAFB2*	9667	24,894	ref (0), I (2), II (0), III (3), SCC (3)
chr19:5598484–5623377	*SAFB*	6294	24,894	ref (0), I (2), II (0), III (3), SCC (3)
chr19:5623377–5625242	*SAFB*	6294	1,866	ref (0), I (2), II (0), III (3), SCC (3)
chr19:5625242–5625729	*SAFB*	6294	488	ref (0), I (2), II (0), III (3), SCC (3)
chr19:5625729–5846816	*SAFB*	6294	221,088	ref (0), I (2), II (0), III (3), SCC (3)

Reference (0), I (1), II (0), III (3), SCC (3) means a 361-kb deletion on chromosome 1 occurred in one subject with CIN-I, 3 subjects with CIN-III and 3 subjects with SCC.

**Table 6 medsci-04-00012-t006:** Functional enrichment of Kyoto Encyclopedia of Genes and Genomes (KEGG) pathway analysis in the identified CNAs (>100 kb and occurring in two or more patients with SCC).

Pathway Name	Enrichment Score	Enrichment *p* Value	% Genes in Pathway	# Gene in List	KEGG Pathway ID
Insulin signaling pathway	4.749	0.009	7.03	9	16
Endometrial cancer	4.698	0.009	10.64	5	11
Parkinson’s disease	4.054	0.017	7.22	7	215
Glioma	3.713	0.024	8.33	5	182
Huntington’s disease	3.640	0.026	5.84	9	52
NF-κB signaling	3.572	0.028	7.14	6	163
Ribosome	3.522	0.030	7.06	6	275
Alcoholism	3.495	0.030	5.70	9	191
Neurotrophin signaling pathway	3.399	0.033	6.31	7	50
Estrogen signaling pathway	3.326	0.036	6.75	6	154
RNA polymerase	3.195	0.041	10.71	3	238
Dopaminergic synapse	3.042	0.048	5.83	7	20
PI3K-Akt signaling pathway	3.038	0.048	4.52	14	262
Ubiquitin-mediated proteolysis	3.005	0.049	5.79	7	175

# ≥6 genes in each function were listed here.

**Table 7 medsci-04-00012-t007:** Gene ontology (GO) categories of CNAs (>100 kb and occurring in two or more than two patients with SCC).

Type	Function	Enrichment Score	Enrichment *p* Value	% Genes in Group	# Genes in List	GO ID
Molecular function	signal transducer activity	5.344	0.005	0.477	7	4871
	molecular transducer activity	4.780	0.008	0.430	7	60089
	enzyme binding	3.089	0.046	0.376	5	19899
	protein kinase binding	5.858	0.003	0.998	4	19901
	kinase binding	5.405	0.004	0.879	4	19900
Cellular component	membrane-bounded vesicle	4.420	0.012	0.338	9	31988
	vesicle	4.193	0.015	0.326	9	31982
	plasma membrane	3.061	0.047	0.271	9	5886
	plasma membrane part	3.388	0.034	0.328	7	44459
	plasma membrane protein complex	5.500	0.004	0.903	4	98797
	membrane protein complex	3.203	0.041	0.457	4	98796
Biological process	signal transduction	4.648	0.010	0.291	12	7165
	cell surface receptor signaling pathway	4.546	0.011	0.344	9	7166
	regulation of signal transduction	3.443	0.032	0.332	7	9966
	positive regulation of gene expression	3.992	0.018	0.413	6	10628
	macromolecular complex subunit organization	3.303	0.037	0.354	6	43933
	G-protein coupled receptor signaling pathway	4.089	0.017	0.489	5	7186
	negative regulation of response to stimulus	3.821	0.022	0.457	5	48585
	defense response	3.783	0.023	0.452	5	6952
	positive regulation of protein metabolic process	3.569	0.028	0.428	5	51247

# ≥6 genes in each function were listed here.

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
