# Peer review of "Novel Somatic Copy Number Alteration Identified for Cervical Cancer in the Mexican American Population"

_medsci, 2016, doi:10.3390/medsci4030012_

Round 1
Reviewer 1 Report
Tobari and colleagues studied the CNV
burden in patients with cervical cancer from Mexican American
ancestry. They found that the burden of deletions in these patients
is much higher than expected, as compared with CNV burden in HapMap
populations. They observed that their sample is enriched in CNV
variants in pathways relevant to the phenotype like endometrial
cancer and estrogen signaling pathway. I think that these finding
are indeed intriguing and deserve to be pursued in a larger study.
However, I think that the manuscript needs some substantial revisions
before it can be published.
Major comments:
1-My main issue is with the way the results are presented. The authors claim that they observed statistical differences between case and reference groups yet they do not preform a statistical test, nor give any p-value. Even if they performed a test, important questions remain on the selection of the reference. The ancestry of HapMap is not latino so therefore any test between cases and references will be confounded by ancestry. The PCA they show is probably capturing those differences instead of real differences given by disease status and, by the way, I question the usefulness of this PCA. My suggestion is to offer a better background for these results by trying to answer two main questions: 1) is the deletion burden observed really exceptional in a population sample of similar ancestry? To answer this question, I would refer to the 1000 genomes data where there are Mexicans, Colombians, Puerto Ricans and Peruvians. There must be available CNV estimations for these ancestries. If you find good estimates then perhaps a statistical tests with those as your control group would be more believable. 2) is the deletion burden observed similar to those for other cancers? Then again there must be estimates of deletion burden for different types of cancers from TCGA studies. Please cite those.
2- The only real statistical inference they show is on the pathway analysis yet no mention is made on the abstract. I suggest bringing this result up to the surface and making it more relevant even from the abstract. You can refer to it as methodological alternative, in the introduction, for the limited number of cases and lack of control group; issues that must be tackle in a larger study.
Minor comments:
1-I missed a definition of the sample's ancestry. In the title it refers to Mexican American but in the rest of the manuscript it is treated as a latino population. Note that there are differences in admixture in the latino population depending of their country of origin. Studies on the 1000 genomes show differences between Puerto Ricans, Colombians, Mexicans and Peruvians. Therefore, if the authors can state that their sample is essentially Mexican that would be more informative and precise. I would then limit the use of latino. However if the ancestry is only self reported as latino, then I would write latino in the title and mention the issue of genetic variability between latinos in the discussion.
2-Abstract Line 23: should read Latino Populations
3-Figure 1: I cannot see the supposed deletions marked in green
4-Table 2b: Please remove the % sign in some of the table cells, for instance in Reference (Deletion, 10-100kb).
Author Response
1-My main issue is with the way the results are presented. The authors claim that they observed statistical differences between case and reference groups yet they do not preform a statistical test, nor give any p-value. Even if they performed a test, important questions remain on the selection of the reference. The ancestry of HapMap is not latino so therefore any test between cases and references will be confounded by ancestry. The PCA they show is probably capturing those differences instead of real differences given by disease status and, by the way, I question the usefulness of this PCA. My suggestion is to offer a better background for these results by trying to answer two main questions: 1) is the deletion burden observed really exceptional in a population sample of similar ancestry? To answer this question, I would refer to the 1000 genomes data where there are Mexicans, Colombians, Puerto Ricans and Peruvians. There must be available CNV estimations for these ancestries. If you find good estimates then perhaps a statistical tests with those as your control group would be more believable. 2) is the deletion burden observed similar to those for other cancers? Then again there must be estimates of deletion burden for different types of cancers from TCGA studies. Please cite those.
[Response] thank you for reviewer’s comments, we made changes accordingly. We added statistical analysis for CNA association between case and reference groups (please see page 5).
1) For the questions of “is the deletion burden observed really exceptional in a population sample of similar ancestry”, we agree with the reviewer’s comment. We did the analysis a couple of years ago, at that time, there is no CNV data from the 1000 genome for Mexicans, Colombians, Puerto Ricans and Peruvians, however, there is data for the admixed populations, thus we used this for this study. There might be some difficulty to do more analysis since our statistician and bioinformatics person left the department and we are in the process of hiring somebody for this role. At the same, we continue in recruiting more subjects with cervical cancers to further confirm our findings. Moreover, in the discussion, we have list this as one of the limitations in page 8 “2) there might be bias of the CNA identified in cervical cancer for the Mexican Americans since we used the 1000 Genome admixed populations, not Mexican Americans, thus, we current recruit more subjects with cervical cancer from the same population and plan to validate the findings in more samples;”
2) For the question of “is the deletion burden observed similar to those for other cancers? Then again there must be estimates of deletion burden for different types of cancers from TCGA studies”, our response is that we did search of CNAs observed in other cancer based on TCGA studies, although there are limited studies, we addressed this issue in the discussion section (page 8) “…Using the TCGA data, a recent study identified nine regions of deletion that were unique to ER+ post menopause tumors in patients with breast cancer, including deletion in 7p22.3 where our newly identified deletion in cases only located and it contains known tumor suppressor gene”.
2- The only real statistical inference they show is on the pathway analysis yet no mention is made on the abstract. I suggest bringing this result up to the surface and making it more relevant even from the abstract. You can refer to it as methodological alternative, in the introduction, for the limited number of cases and lack of control group; issues that must be tackle in a larger study.
[Response] great suggestions, thank the reviewer.
Now we added a sentence of “Moreover, the pathway analysis revealed endometrial cancer and estrogen signaling pathways associated with this cancer (P < 0.05) using the KEGG” in the abstract as the reviewer’s suggestion. In addition, in the introduction section, we added “We are aware of the limited number of cases and lack of control group. Thus future a large study with a control sample and more cases as methodological alternative is needed”. (Please see page 3)
Minor comments:
1-I missed a definition of the sample's ancestry. In the title it refers to Mexican American but in the rest of the manuscript it is treated as a latino population. Note that there are differences in admixture in the latino population depending of their country of origin. Studies on the 1000 genomes show differences between Puerto Ricans, Colombians, Mexicans and Peruvians. Therefore, if the authors can state that their sample is essentially Mexican that would be more informative and precise. I would then limit the use of latino. However if the ancestry is only self reported as latino, then I would write latino in the title and mention the issue of genetic variability between latinos in the discussion.
[Response] as the reviewer’s suggestion, we changed “Latino” to “Mexican American” in the text of the manuscript and table
2-Abstract Line 23: should read Latino Populations
[Response] now we change “Latino population” to “Mexican American” as the reviewer’s suggestion
3-Figure 1: I cannot see the supposed deletions marked in green
[Response] thank you for the comments. We also realized this issue which is due to that the color green overlaps with the other colors, thus it cannot be distinguished, and we have tried different color combinations without help. Now we made changes of the figure and legend.
4-Table 2b: Please remove the % sign in some of the table cells, for instance in Reference (Deletion, 10-100kb).
[Response] thanks, we removed the % sign in table cells in Table 2b
Reviewer 2 Report
A brief summary
The authors conducted a pilot study of genome wide CNA analysis using 2.5 million markers in four diagnostic groups: reference, low grade dysplasia, high grade dysplasia, and invasive carcinoma. They found statistically significant difference of CNA burden between case and reference groups in different sizes of CNAs that included deletions and amplifications, e.g., a statistically significant difference of >100 kb deletions were observed between the reference and pre-cancer and cancer groups. Recurrent aberrations of 98 CNA regions were also identified in cases only. A total of 32 CNA regions identified contained tumor suppressor genes and oncogenes.
Broad comments
This is the first genome-wide copy number analysis of CNAs identified for cervical cancer in the U.S. Latino population. The manuscript is well-written. The experimental design is adequate and the statistical methods are updated. The results are novel and provide insight into how multiple CNAs might contribute to cervical cancer development. No major issues were found.
The authors have acknowledged the limitation of this study, for example, the small sample size. Therefore, the authors pointed out that validation and confirmation of the results in a large sample size will be needed in the future.
Specific comments
There comments are minor.
1) Line 34. The authors need to make clear about “pre-, cancer (91.3%) groups”. It is not clear if there is one or two groups but just one percentage.
2) Line 40. The authors may add up to 6 key words.
3) Lines 196-197. It is not clear which statistical method is used for the significant results in Table 2.Please give more details.
4) Line 199. The authors need to define “pre-cancer” in materials.
5) Lines 202-206. The authors mentioned the results by using PCA but PCA was not found in methods. The authors need to give more details about how to use PCA in statistical methods.
6) Lines 214-216. It is not clear which statistical method is used for the significant results. Please give more details.
7) Lines 238-242 are replicated in discussion lines 376-380. The authors may need to delete the part in lines 238-242.
Author Response
There comments are minor.
1) Line 34. The authors need to make clear about “pre-, cancer (91.3%) groups”. It is not clear if there is one or two groups but just one percentage.
[Response] thank you, reviewer, this is a typo, we corrected it.
2) Line 40. The authors may add up to 6 key words.
[Response] yes, we added up to six key words now
3) Lines 196-197. It is not clear which statistical method is used for the significant results in Table 2. Please give more details.
[Response] we made changes based on the reviewer’s suggestion, please see statistical method in page 5 and result in page 6
4) Line 199. The authors need to define “pre-cancer” in materials.
[Response] thanks, we provided definition for pre-cancer in materials, which include patients with CIN I, II, and III.
5) Lines 202-206. The authors mentioned the results by using PCA but PCA was not found in methods. The authors need to give more details about how to use PCA in statistical methods.
[Response]
6) Lines 214-216. It is not clear which statistical method is used for the significant results. Please give more details.
[Response] thanks, we added “…using statistical analyses described in the method section”
7) Lines 238-242 are replicated in discussion lines 376-380. The authors may need to delete the part in lines 238-242.
[Response] thanks, the reviewer. Now we removed redundancy statement.
Round 2
Reviewer 1 Report
I have no further comments